# Hi-Time: Hierarchical Latent Prediction for Multivariate Time Series Classification

Kun Zeng [1]   Binquan Wu [1]   Qianli Ma [1] [†]

## Abstract

Integrating Large Language Models (LLMs) into time series tasks has yielded impressive performance. While some works aim to enhance accuracy by explicitly designing step-by-step reasoning into prompts, such explicit Chain-of-Thought (CoT) approaches are difficult to generalize to time series. This is because it is difficult to clearly define the reasoning trajectories of time series. In addition, the high heterogeneity across time series often requires specialized prompt designs, limiting the model's scalability. To address these challenges, we propose **Hi-Time**, a **hi**erarchical latent prediction framework based on temporal semantic codes for multivariate **time** series classification. This framework automatically constructs scenario-specific coarse-to-fine prediction trajectories based on the characteristics of time series, thereby providing structured supervision for the LLM. Specifically, Hi-Time first performs temporal representation pre-training with a multi-view temporal representation fusion to acquire high-quality temporal embeddings. We then discretize these temporal embeddings into hierarchical temporal semantic codes that form the coarse-to-fine prediction trajectory. Finally, the LLM predicts temporal semantic codes in a stepwise manner and then infers the final label, thereby establishing a coarse-to-fine decision process. Experiments on ten public multivariate time series datasets demonstrate that Hi-Time effectively adapts to diverse datasets and outperforms state-of-the-art methods. Our code is available at https://github.com/qianlima-lab/Hi-Time.

---

[1]School of Computer Science and Engineering, South China University of Technology, Guangzhou, China. Correspondence to: Qianli Ma <qianlima@scut.edu.cn>.

*Proceedings of the 43rd International Conference on Machine Learning*, Seoul, South Korea. PMLR 306, 2026. Copyright 2026 by the author(s).

## 1. Introduction

Multivariate time series classification (MTSC) serves as a cornerstone for a wide range of critical real-world applications, including human activity recognition (Yang et al., 2015), industrial fault diagnosis (Liu et al., 2016), and clinical analysis (Song et al., 2018). Over the past decade, the field has undergone a significant transformation driven by deep neural networks. A diverse set of architectures, ranging from Convolutional Neural Networks (CNNs) (Bai, 2018) and Recurrent Neural Networks (RNNs) (Siami-Namini & Namin, 2018) to Transformer-based models (Liu et al., 2021; Zhou et al., 2021; Wu et al., 2022; Zhou et al., 2022; Woo et al., 2022; Kitaev et al., 2020; Nie, 2022), has been developed, consistently pushing the boundaries of state-of-the-art performance.

More recently, the rise of foundation models has driven significant breakthroughs in Natural Language Processing (NLP) and Computer Vision (CV). Models like GPT-4 (Achiam et al., 2023) and SAM (Kirillov et al., 2023) demonstrate the powerful generalization capabilities by large-scale pre-training. Inspired by these successes, researchers seek to harness the reasoning abilities and extensive world knowledge of LLMs to tackle complex challenges in time-series analysis. A central focus of this effort is bridging the modality gap between continuous time series and discrete text. Existing solutions typically follow two main strategies: (1) converting time series data into textual descriptions and performing zero-shot inference via prompt engineering (Gruver et al., 2023; Xue & Salim, 2023; Cao et al., 2023; Zhao et al., 2025; Lee et al., 2025; Tao et al., 2024), or (2) introducing learnable reprogramming layers that project temporal embeddings into the LLM's latent space (Jin et al., 2023; Chang et al., 2023; Zheng et al., 2025). However, relying solely on converting data into textual descriptions or on reprogramming layers is often insufficient to fully elicit LLMs' temporal capabilities over time series.

Consequently, within the paradigm of textualization and prompt engineering, attempts (Xue & Salim, 2023; Cao et al., 2023; Tao et al., 2024) have been made to leverage explicit CoT prompting to guide the model to reason step by step. Yet hand-designed reasoning proves inherently prob-

lematic in the time series. First, the semantic ineffability of time series: unlike natural language, which is governed by discrete logical syntax, time series patterns are continuous, abstract, and often noisy. Converting complex signal fluctuations into an accurate, step-by-step written reasoning process is usually very difficult, even for domain experts. Second, the distributional heterogeneity across domains: time series from disparate fields (e.g., finance vs. healthcare) exhibit vastly different characteristics. Therefore, the inherent difficulty of explicitly defining correct reasoning trajectories for diverse time series severely limits the scalability and transferability of these prompt-based frameworks.

To address the semantic ineffability of explicitly defining reasoning paths and the rigidity of handcrafted templates, we propose a novel framework, **Hi-Time** (**Hi**erarchical Latent Prediction for **Time** Series Classification). Rather than relying on manually designed textual prompts, Hi-Time enables the model to autonomously construct a coarse-to-fine prediction trajectory directly from the temporal patterns and dependencies in the data. First, we conduct temporal pre-training, during which we employ a multi-view representation fusion as a core component to learn high-quality temporal semantic representations by integrating information from three key perspectives: multi-scale fusion, cross-variable fusion, and temporal dynamics fusion. Second, we apply hierarchical residual quantization to the temporal representations, transforming continuous features into a sequence of hierarchical temporal semantic codes that capture a coarse-to-fine semantic trajectory. Finally, we introduce a hierarchical latent prediction strategy that leverages the pretrained LLM's capability. Instead of directly mapping the final-layer representation to labels, we use the LLM's final-layer representation to predict temporal semantic codes progressively. At each step, the original temporal representation is concatenated with the embeddings of all previously predicted codes to produce the next-level code. Once all code levels have been generated, the final label is inferred conditioned on the complete sequence of semantic embeddings. This formulation enforces a coarse-to-fine prediction process: earlier codes capture high-level temporal semantics, while later codes progressively refine finer-grained details. This structure provides structured supervision for the LLM, thereby improving temporal classification accuracy.

In summary, the main contributions of this paper are as follows:

- We propose Hi-Time, a framework that replaces rigid handcrafted CoT prompts with autonomously learned coarse-to-fine prediction trajectories. By internalizing this prediction process in the latent space, it overcomes the semantic ineffability and distributional heterogeneity of textual prompts in time-series classification.

- We introduce a multi-view representation fusion that,

through temporal pre-training, learns high-quality temporal representations by jointly integrating multi-scale, cross-variable, and temporal dynamics information.

- We conduct extensive experiments across 10 diverse time series benchmarks, demonstrating the superior performance of Hi-Time over state-of-the-art methods and its strong adaptability to diverse data distributions.

## 2. Related work

**Multivariate Time Series Feature Modelling.** Modelling the intrinsic characteristics of complex temporal dynamics is pivotal for practical multivariate time series analysis. Contemporary deep learning approaches typically address this by incorporating specific inductive biases, which generally crystallise into three primary paradigms: multi-scale modelling, cross-variable modelling, and temporal dynamics modelling. Multi-scale modelling targets signal variations across diverse resolutions; architectures (Wang et al., 2024b;a; Liu et al., 2021; Wu et al., 2022) such as PyraFormer (Liu et al., 2021) and TimesNet (Wu et al., 2022) decompose complex patterns into multi-resolution representations to capture inherent hierarchies and periodicities. Cross-variable modelling prioritises spatial correlations among sensors or channels in multivariate data. Recent innovations such as iTransformer (Liu et al., 2023) and Crossformer (Zhang & Yan, 2023) explicitly aggregate multivariate information to characterise inter-variable dependencies. Meanwhile, temporal dynamics modelling centres on discerning the varying importance of local temporal segments. SoftShape (Liu et al.) introduces a soft sparsification mechanism that weighs temporal subsequences based on their classification contribution scores. However, existing approaches primarily treat these dimensions independently or only partially combine them, lacking a unified framework that integrates multi-scale, cross-variable, and temporal dynamic modelling.

**Large Language Models for Time Series.** Large Language Models (LLMs) have demonstrated powerful reasoning and generalization abilities, driving significant advancements in domains like Computer Vision (Wu et al., 2024; Lu et al., 2024) and Recommender Systems (Geng et al., 2022; Rajput et al., 2023). Consequently, a growing body of work is now dedicated to adapting these foundational models to the domain of time series. Pioneering approaches, such as PromptCast (Xue & Salim, 2023) and LLMTime (Gruver et al., 2023), recast time series forecasting as a language modelling problem by serializing continuous values into numerical text tokens, enabling zero-shot inference with models like GPT-4. To address the inherent modality gap between continuous signals and discrete text, subsequent frameworks, including OFA (Zhou et al., 2023), LLM4TS (Chang et al., 2023), Time-LLM (Jin et al., 2023),

MAP4TS (Lee et al., 2025), and InstructTime (Cheng et al., 2025), adopt alignment mechanisms such as patch-based alignment and neural reprogramming. These methods map temporal embeddings directly into the LLM's latent input space. More recently, studies (Xue & Salim, 2023; Cao et al., 2023) have attempted to integrate explicit Chain-of-Thought (CoT) prompts to guide models. However, transposing explicit CoT to the temporal domain presents fundamental challenges. Unlike natural language, the reasoning trajectory of time series is semantically ambiguous and intractable to articulate in textual form. The inherent difficulty of explicitly defining correct reasoning trajectories for diverse time series severely constrains the scalability and generalization of such frameworks.

**Representation Discretization.** Discretising continuous data into semantic tokens serves as a critical nexus between high-dimensional signals and symbolic representations. Seminal works in Computer Vision (Van Den Oord et al., 2017; Bao et al., 2021) and Recommender Systems (Rajput et al., 2023) have already demonstrated that discrete latent spaces, particularly those derived via Residual Quantisation, effectively encapsulate high-level semantic abstractions. In the temporal domain, contemporary approaches leverage vector quantization primarily for generation (Lee et al., 2023), long-term forecasting (Feng et al., 2025), or modality alignment (Cheng et al., 2025). However, these methodologies predominantly treat discrete tokens as intermediate representations for data compression, reconstruction, or static alignment. The potential of hierarchical discrete codes to function as a coarse-to-fine prediction trajectory that progressively guides downstream classification remains largely unexplored.

## 3. Preliminaries

### 3.1. Problem Definition

In this study, we focus on multivariate time series classification. Let the time series dataset be represented as $\mathcal{D} = \{(\mathcal{X}_n, \mathcal{Y}_n)\}_{n=1}^{N}$, where $N$ denotes the total number of time series samples. Each multivariate time series $\mathcal{X}_n \in \mathbb{R}^{T \times D}$ consists of an ordered sequence of $T$ time steps, denoted as $\mathcal{X}_n = \{\mathbf{x}_1, \mathbf{x}_2, \ldots, \mathbf{x}_T\}$, where each observation $\mathbf{x}_t \in \mathbb{R}^D$ represents a vector of $D$ recorded variables. The corresponding label $\mathcal{Y}_n \in \{0, 1\}^C$ is a one-hot encoded vector, where each element $y_c \in \{0, 1\}$ indicates whether $\mathcal{X}_n$ belongs to class $c$, with $C$ representing the total number of classes. Given a deep learning model parameterized by $\theta$, the objective of the task is to optimize the function $f_\theta : \mathbb{R}^{T \times D} \to \mathbb{R}^C$ such that it can accurately predict the label $\mathcal{Y}_n$ corresponding to any input time series $\mathcal{X}_n$.

## 4. Methodology

The overall architecture of Hi-Time, illustrated in Figure 1, comprises three phases: **temporal representation pre-training**, **temporal semantic codes generation**, and **hierarchical latent prediction**. In the temporal representation pre-training phase, a multi-view representation fusion is used to learn unified temporal semantic embeddings from raw multivariate time series. It jointly performs multi-scale, cross-variable, and temporal dynamics fusion (see Section 4.1). In the temporal semantic code generation phase, these continuous embeddings are discretised via hierarchical residual quantisation into a sequence of temporal semantic codes. This produces a coarse-to-fine prediction trajectory from the underlying temporal structure (see Section 4.2). In the hierarchical latent prediction phase, the LLM progressively predicts temporal semantic codes. At each step, the original temporal representation is concatenated with the embeddings of all previously predicted temporal semantic codes to produce the next-level code. Once all code levels have been generated, the final label is inferred conditioned on the complete sequence of semantic embeddings (see Section 4.3).

### 4.1. Temporal Representation Pre-training

This phase aims to learn unified temporal semantic representations from raw multivariate time series, which provide the continuous temporal features to be discretized in the next stage. To capture complementary structural patterns in temporal data, we introduce a multi-view representation fusion that sequentially performs three fusion operations: multi-scale fusion, cross-variable fusion, and temporal dynamics fusion.

Given an input time series $\mathbf{X} \in \mathbb{R}^{T \times D}$, where $D$ is the number of variables and $T$ is the sequence length, we first adopt PatchTST (Nie, 2022) as the backbone to obtain patch-level hidden states $\mathcal{H} \in \mathbb{R}^{D \times N \times d_{\text{model}}}$, where $N$ is the number of patches. The representation $\mathcal{H}$ is then passed through the three fusion modules sequentially, yielding the final semantic embedding $z_{\text{expert}} \in \mathbb{R}^{d_{\text{model}}}$ for downstream temporal semantic code generation.

#### 4.1.1. MULTI-SCALE FUSION

Temporal patterns may exhibit different characteristics at different resolutions, and these diverse forms can provide complementary information. To capture such complementary information, we construct a multi-scale set $\{\mathbf{X}^{(m)}\}_{m=1}^{M}$ by adaptively downsampling the original series $\mathbf{X} \in \mathbb{R}^{T \times D}$ along the temporal dimension.

Concretely, we first apply the Fast Fourier Transform (FFT) to $\mathbf{X}$ to obtain its frequency spectrum and identify the top-$S$ dominant frequencies $\{f_s\}_{s=1}^{S}$ with the largest amplitudes. These dominant frequencies are then used to derive a set of

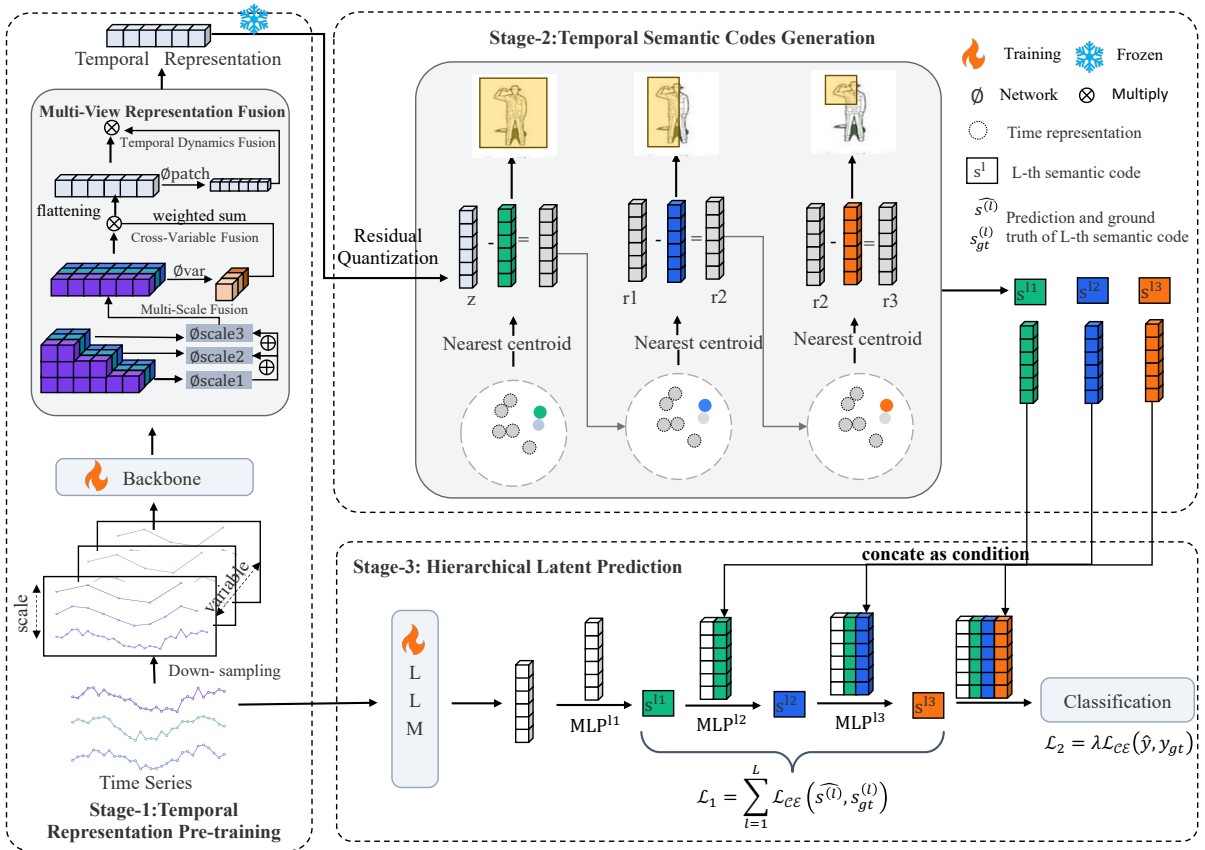

*Figure 1.* The overall architecture of the proposed Hi-Time framework, which comprises three key phases: (1) **Temporal Representation Pre-training**, where a multi-view representation fusion integrates multi-scale, cross-variable, and temporal dynamics information to learn unified temporal semantic embeddings; (2) **Temporal Semantic Codes Generation**, which utilizes hierarchical residual quantization to discretize continuous features into a sequence of coarse-to-fine semantic codes serving as the prediction trajectory, which is similar to human action recognition: we first observe the whole body, then focus on the left side, and finally zoom in on the upper-left part to refine the remaining details; and (3) **Hierarchical Latent Prediction**, where the LLM progressively predicts these semantic codes step by step to form the prediction trajectory before inferring the final classification label.

downsampling steps $\{p_m\}_{m=1}^M$, where each $p_m$ is chosen to be inversely proportional to the corresponding dominant frequency, i.e., $p_m \propto \frac{1}{f_{s(m)}}$. For each scale $m$, we downsample $\mathbf{X}$ with step size $p_m$ to obtain

$$\mathbf{X}^{(m)} = \text{Downsample}(\mathbf{X}; p_m) \in \mathbb{R}^{T_m \times D}, \qquad (1)$$

where $T_m = \left\lfloor \frac{T}{p_m} \right\rfloor$ denotes the length of the downsampled sequence at scale $m$. Each scale is then encoded by the same backbone, yielding scale-specific hidden states $\mathcal{H}^{(m)} \in \mathbb{R}^{D \times N \times d_{\text{model}}}$.

We fuse multi-scale information via additive aggregation. Specifically, for each scale $m$, we apply a scale-specific MLP to obtain a scale embedding, and then sum embeddings

across scales:

$$h^{(m)} = \text{MLP}_s^{(m)}\left(\mathcal{H}^{(m)}\right) \in \mathbb{R}^{D \times N \times d_{\text{model}}}, \qquad (2)$$

$$z_{\text{scale}} = \sum_{m=1}^M h^{(m)} \in \mathbb{R}^{D \times N \times d_{\text{model}}}. \qquad (3)$$

### 4.1.2. CROSS-VARIABLE FUSION

Multivariate time series often exhibit strong inter-variable dependencies, which are crucial for reliable representation learning. To explicitly model such correlations, we perform cross-variable fusion by treating the $D$ variables as tokens and applying self-attention along the variable dimension independently for each patch.

Given the multi-scale fused features $z_{\text{scale}} \in \mathbb{R}^{D \times N \times d_{\text{model}}}$, we model inter-variable dependencies by applying self-attention over the variable dimension for each patch $n \in$

$\{1, \ldots, N\}$:

$$[Q^{(n)}, K^{(n)}, V^{(n)}] = z_{\text{scale}}^{(n)}[W_q, W_k, W_v], \qquad (4)$$

$$A_{\text{var}}^{(n)} = \text{Softmax}\left(\frac{Q^{(n)} K^{(n)\top}}{\sqrt{d_k}}\right) \in \mathbb{R}^{D \times D}, \qquad (5)$$

$$z_{\text{var}}^{(n)} = A_{\text{var}}^{(n)} V^{(n)} + z_{\text{scale}}^{(n)}, \qquad (6)$$

where $z_{\text{scale}}^{(n)} = z_{\text{scale}}[:, n, :] \in \mathbb{R}^{D \times d_{\text{model}}}$ denotes the features at patch position $n$.

### 4.1.3. TEMPORAL DYNAMICS FUSION

Temporal patches are not equally informative: different segments may contribute differently to the classification decision. To capture such patch-level importance, we perform adaptive fusion over the patch and variable dimensions on top of the cross-variable representation.

Let $z_{\text{var}} \in \mathbb{R}^{D \times N \times d_{\text{model}}}$ denote the cross-variable fused representations. We compute gated importance scores for each patch through a two-layer MLP:

$$g_{d,n} = W_2 \cdot \text{GELU}(W_1 \cdot z_{\text{var}}^{(d,n)}) \in \mathbb{R}, \qquad (7)$$

$$\alpha_{d,n} = \frac{\exp(g_{d,n})}{\sum_{n'=1}^{N} \exp(g_{d,n'})}, \qquad (8)$$

where $W_1 \in \mathbb{R}^{d_{\text{model}}/2 \times d_{\text{model}}}$ and $W_2 \in \mathbb{R}^{1 \times d_{\text{model}}/2}$ are learnable parameters. The final representation is obtained by weighted aggregation over patches, followed by averaging across variables:

$$z_{\text{expert}} = \frac{1}{D} \sum_{d=1}^{D} \sum_{n=1}^{N} \alpha_{d,n} \, z_{\text{var}}^{(d,n)} \in \mathbb{R}^{d_{\text{model}}}, \qquad (9)$$

The resulting $z_{\text{expert}}$ highlights temporally salient patches while suppressing less informative ones. We pre-train the encoder using the same classification objective to obtain discriminative representations. After pre-training, the encoder and the multi-view representation fusion are frozen, and the learned representation $z_{\text{expert}}$ serves as input for the subsequent hierarchical temporal semantic code generation stage.

### 4.2. Temporal Semantic Codes Generation

To provide the LLM with a structured coarse-to-fine supervision signal, we discretize the unified embedding $z_{\text{expert}}$ into a sequence of temporal semantic codes. Instead of using a flat quantization, we employ **Residual Quantization** to decompose the continuous temporal representation into hierarchical temporal semantic codes. This approach effectively translates abstract temporal features into an ordered coarse-to-fine trajectory, where earlier codes capture coarse patterns, and later codes refine them with increasingly fine-grained details.

We implement this process via Residual Quantization K-Means (RQ-KMeans), utilizing $L$ quantization stages to generate a target code sequence $S_{\text{gt}} = (s^{(1)}, s^{(2)}, \ldots, s^{(L)})$. Formally, let $\{\mathcal{C}^{(l)}\}_{l=1}^{L}$ denote a set of learnable codebooks, where each codebook $\mathcal{C}^{(l)} = \{e_k^{(l)}\}_{k=1}^{K}$ consists of $K$ prototype vectors corresponding to the $l$-th level of granularity.

The quantization is performed hierarchically. At the first stage ($l = 1$), we approximate the semantic representation $z_{\text{expert}}$ by identifying the nearest prototype in the first codebook, capturing the coarsest temporal semantics:

$$s^{(1)} = \arg\min_k \left\| z_{\text{expert}} - e_k^{(1)} \right\|_2. \qquad (10)$$

For subsequent stages $l > 1$, we compute the residual vector by subtracting the selected embedding from the previous stage and map it to the nearest prototype in the $l$-th codebook:

$$r^{(l)} = r^{(l-1)} - e_{s^{(l-1)}}^{(l-1)}, \quad r^{(0)} = z_{\text{expert}}, \qquad (11)$$

$$s^{(l)} = \arg\min_k \left\| r^{(l)} - e_k^{(l)} \right\|_2. \qquad (12)$$

Here, $e_{s^{(l)}}^{(l)}$ denotes the selected embedding from stage $l$. This recursive process yields a hierarchical sequence of temporal semantic codes $S_{\text{gt}}$, which serves as the ground-truth prediction trajectory for the subsequent hierarchical latent prediction phase.

### 4.3. Hierarchical Latent Prediction

In this phase, we leverage an LLM to perform stepwise prediction in the latent space. Rather than directly predicting the class label from the input, we train the model to sequentially infer the hierarchical semantic codes $S_{\text{gt}}$ and then generate the final decision conditioned on the resulting code trajectory. In our implementation, the LLM is instantiated as GPT-2 (Radford et al., 2019).

### 4.3.1. INPUT EMBEDDING

To align continuous-time series data with the LLM's semantic space, we use a learnable linear projection layer rather than converting the data into natural language tokens. Given the input time series $\mathbf{X}$, we map it into the model's embedding dimension $d_{\text{llm}}$:

$$\mathbf{E} = \text{Linear}(\mathbf{X}) + \mathbf{P}_{\text{pos}}, \qquad (13)$$

where $\text{Linear}(\cdot)$ denotes a linear transformation layer, and $\mathbf{P}_{\text{pos}}$ represents the positional embeddings required to retain temporal order information. The resulting embeddings $\mathbf{E}$ are then fed into the LLM backbone.

### 4.3.2. MULTI-STEP PREDICTION

Let $H \in \mathbb{R}^{T \times d_{\text{llm}}}$ be the last-layer hidden states of the LLM encoded from $\mathbf{E}$, where $T$ is the input sequence length.

We use the last token representation $h_{\text{last}} \in \mathbb{R}^{d_{\text{llm}}}$ as the global temporal context. We represent each semantic code $s^{(l)}$ with an embedding lookup $g^{(l)} = \text{Emb}^{(l)}(s^{(l)})$. At step $l$, we predict the next code using an MLP conditioned on the LLM representation and the sequence of previously predicted codes, denoted as $g^{(1:l-1)}$:

$$\hat{s}^{(l)} = \text{MLP}_{\text{code}}^{(l)}\left(\left[h_{\text{cls}} \| g^{(1:l-1)}\right]\right), \quad l = 1, \ldots, L. \quad (14)$$

This formulation enforces a coarse-to-fine prediction process in which earlier codes capture high-level temporal semantics and later codes refine details.

After obtaining the full code embeddings $g^{(1:L)}$, we form the final prediction by concatenating the LLM representation with $g^{(1:L)}$:

$$\hat{y} = \text{MLP}_{\text{cls}}\left(\left[h_{\text{cls}} \| g^{(1:L)}\right]\right). \quad (15)$$

### 4.3.3. TRAINING OF HI-TIME

To enable effective hierarchical latent prediction, the training objective is designed to jointly supervise the prediction of the hierarchical semantic trajectory and the final decision-making. Specifically, the total loss function $\mathcal{L}$ combines the step-wise code prediction error with the classification error:

$$\mathcal{L} = \sum_{l=1}^{L} \mathcal{L}_{\text{CE}}\left(\hat{s}^{(l)}, s_{\text{gt}}^{(l)}\right) + \lambda \, \mathcal{L}_{\text{CE}}(\hat{y}, y_{\text{gt}}), \quad (16)$$

where $\lambda$ is a hyperparameter to adjust the training loss ratio.

## 5. Experiments

We demonstrate Hi-Time's effectiveness in time-series classification across diverse datasets. Ablation studies confirm the contributions of its multi-view representation fusion and the hierarchical latent prediction enabled by temporal semantic codes, and further demonstrate its robustness through comprehensive hyper-parameter analyses. Finally, visualisations illustrate the meaningful hierarchical semantics captured by temporal semantic codes.

### 5.1. Experimental Setup

**Datasets.** We examine our method on ten public MTS datasets for classification, including Human Activity Recognition (HAR) (Anguita et al., 2012), ISRUC (Khalighi et al., 2016), and eight large datasets from UEA archive (Bagnall et al., 2018), i.e., ArticularyWordRecognition (AWR), FingerMovements (FM), SpokenArabicDigitsEq (SAD), CharacterTrajectories (CT), FaceDetection (FD), InsectWingbeat (IW), MotorImagery (MI), and SelfRegulationSCP1 (SRSCP1). Following the evaluation protocol of TS-GAC (Wang et al., 2024c), we adopt the same ten datasets to ensure a fair comparison.

**Baselines.** To evaluate the efficacy of Hi-Time, we benchmark it against a comprehensive suite of competitive baselines spanning diverse modelling paradigms: (1) Self-attention based models TST (Zerveas et al., 2021), PatchTST (Nie, 2022) and FormerTime (Cheng et al., 2023); (2) Convolutional based approaches TCN (Bai, 2018) and MiniROCKET (Dempster et al., 2021); (3) Hybrid model MPTSNet (Mu et al., 2025); (4) Self-supervised learning models TS-TCC (Eldele et al., 2021) and TS-GAC (Wang et al., 2024c); (5) LLM-for-time-series models, including a novel adaptation of GPT-2 (Radford et al., 2019), where its final layer is modified for classification purposes (GPT-As-Classifier), and PromptCast (Xue & Salim, 2023).

**Implementation Details.** We implemented the proposed framework using PyTorch 2.0. To ensure a fair evaluation, we split all datasets into 60%, 20%, and 20% for training, validation, and testing. All experiments were repeated 5 times with different random seeds to verify robustness. For all the baselines except those related to pre-trained language model (PLM), we employed the AdamW optimizer with an initial learning rate of $0.001$ and a weight decay of $1 \times 10^{-4}$, coupled with a Cosine Annealing scheduler. For PLM, we configured the model with the default number of layers and embedding sizes, a pre-training learning rate of $1 \times 10^{-5}$. For Hi-Time, the quantization depth $L$ and codebook size $K$ are independently selected on the validation set of each dataset via grid search over $L \in \{1, 2, 3\}$ and $K \in \{16, 32, 64, 128, 256\}$. All experiments were conducted on a Linux server equipped with a single NVIDIA A800 GPU.

### 5.2. Main Results

Table 1 presents a comprehensive comparison of various models' classification performance over ten datasets. The average ranking (Avg. Rank) reflects the method's relative position based on test accuracy, where lower values signify higher accuracy. Wins indicates the number of datasets where the baseline achieves the highest test accuracy. From the results, it is clear that our Hi-Time model generally outperforms other models across most datasets, achieving the highest overall classification performance. These results indicate the robustness of Hi-Time, which adapts effectively to a variety of classification tasks and diverse data distributions.

Meanwhile, we observe that on several challenging datasets, such as IW, MI, and SRSCP1, Hi-Time achieves notable performance gains, while on others such as FM and FD, our model performs comparably to the strongest baselines. In contrast, improvements on relatively more straightforward datasets are more modest. This trend is consistent with our expectations: for more manageable classification tasks, standard models can already make accurate predictions, so introducing a hierarchical latent prediction process

*Table 1.* Accuracies of Hi-Time and compared methods on ten public datasets. The best results are in **bold**, and the second-best results are underlined.

| Datasets | TST | PatchTST | FormerTime | MPTSNet | TCN | MiniROCKET | TS-TCC | TS-GAC | GPT-2 | PromptCast | **Hi-Time** |
|---|---|---|---|---|---|---|---|---|---|---|---|
| HAR | 0.897 | 0.916 | 0.918 | 0.937 | 0.944 | 0.927 | 0.917 | 0.943 | 0.928 | 0.829 | **0.955** |
| ISRUC | 0.751 | 0.805 | 0.812 | 0.834 | 0.777 | 0.772 | 0.747 | 0.822 | 0.813 | 0.784 | **0.855** |
| AWR | 0.970 | 0.927 | 0.927 | 0.977 | 0.947 | 0.972 | 0.894 | 0.983 | 0.930 | 0.907 | **0.987** |
| FM | 0.580 | 0.580 | 0.598 | **0.640** | 0.440 | 0.628 | 0.474 | 0.520 | 0.524 | 0.540 | **0.640** |
| SAD | 0.974 | 0.977 | 0.970 | 0.954 | 0.964 | 0.993 | 0.952 | 0.980 | 0.959 | 0.968 | **0.995** |
| CT | 0.982 | 0.976 | 0.984 | 0.968 | 0.968 | 0.987 | 0.986 | **0.988** | 0.982 | 0.716 | 0.982 |
| FD | 0.654 | 0.668 | 0.632 | **0.676** | 0.594 | 0.562 | 0.580 | 0.605 | 0.553 | 0.503 | 0.661 |
| IW | 0.674 | 0.540 | 0.543 | 0.584 | 0.600 | 0.418 | 0.561 | 0.658 | 0.553 | 0.515 | **0.695** |
| MI | 0.480 | 0.610 | 0.632 | **0.650** | 0.550 | 0.560 | 0.517 | 0.560 | 0.524 | 0.490 | 0.620 |
| SRSCP1 | 0.860 | 0.795 | 0.887 | 0.898 | 0.781 | 0.863 | 0.836 | 0.845 | 0.823 | 0.577 | **0.908** |
| AVG. Acc | 0.782 | 0.779 | 0.790 | 0.812 | 0.757 | 0.768 | 0.746 | 0.790 | 0.759 | 0.683 | **0.830** |
| Avg. Rank | 6.4 | 6.5 | 5.3 | 3.9 | 7.3 | 5.6 | 8.4 | 4.2 | 7.2 | 9.5 | **2.0** |
| Wins | 0 | 0 | 0 | 3 | 0 | 0 | 0 | 1 | 0 | 0 | **7** |

offers limited additional benefit. For more difficult classification problems, direct label prediction is considerably more complex, and a coarse-to-fine prediction procedure becomes particularly beneficial, enabling the model to handle complex decision boundaries better and improve overall accuracy.

### 5.3. Ablation Study

In this section, we investigate the performance gains contributed by each key component of the proposed method. Specifically, we compare Hi-Time with seven variants: (1) **w/o MVRE**: We remove the entire multi-view representation fusion. (2) **w/o MSF**: We remove the multi-scale fusion module from MVRE. (3) **w/o CVF**: We remove the cross-variable fusion module from MVRE. (4) **w/o TDF**: We remove the temporal dynamics fusion module from MVRE. (5) **w/o MSR**: We remove the hierarchical latent prediction module and directly predict the labels. (6) **w/o Pre-training**: We replace the pre-trained GPT-2 weights with random initialization. (7) **w/o MSL**: We remove the step-wise code prediction loss and supervise only the final classification.

*Table 2.* Ablation study results on selected datasets. The best results are in **bold**. MSL denotes Multi-Step Supervision Loss.

| Method | AWR | IW | Avg. |
|---|---|---|---|
| w/o MVRE | 0.964 | 0.677 | 0.781 |
| w/o MSF | 0.970 | 0.679 | 0.807 |
| w/o MSR | 0.980 | 0.675 | 0.806 |
| w/o TDF | 0.979 | 0.684 | 0.816 |
| w/o CVF | 0.982 | 0.685 | 0.820 |
| w/o Pre-training | 0.952 | 0.658 | 0.784 |
| w/o MSL | 0.976 | 0.680 | 0.814 |
| **Hi-Time (Full)** | **0.987** | **0.695** | **0.830** |

Table 2 shows the ablation results on the AWR and IW datasets. The complete model achieves the highest accuracy,

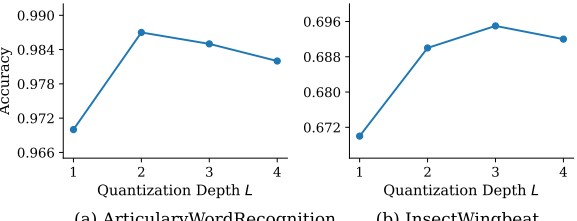

*Figure 2.* Effect of different quantization depth on two datasets.

and removing any key component results in a performance drop, confirming the importance of each design choice in Hi-Time.

Removing the entire **MVRE** causes considerable degradation across datasets, indicating that multi-view representation fusion is necessary for robust representations. Within MVRE, each fusion module is functional: **w/o MSF** leads to a significant drop, while **w/o CVF** and **w/o TDF** cause more minor drops. This shows that joint multi-scale, cross-variable, and temporal dynamics fusion is essential, with multi-scale fusion contributing the most. The hierarchical latent prediction module is also crucial. **w/o MSR** reduces the average accuracy by 2.4%, with a larger drop observed on IW, suggesting that the benefits of hierarchical latent prediction are more pronounced on challenging datasets. In addition, **w/o Pre-training** causes a substantial 4.6% drop, demonstrating that the pre-trained GPT-2 weights provide useful inductive bias beyond what comes from architecture alone. Finally, **w/o MSL** also degrades performance, confirming that the step-wise code prediction loss provides informative gradient signals rather than acting as a redundant auxiliary loss.

### 5.4. Parameter Sensitivity Analysis

**Impact of Quantization Depth $L$.** The number of residual quantization stages $L$ determines the granularity of the

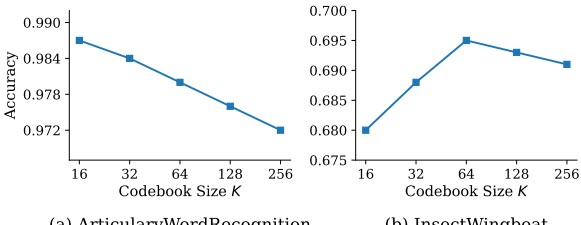

(a) ArticularyWordRecognition    (b) InsectWingbeat

*Figure 3.* Effect of different codebook size on two datasets.

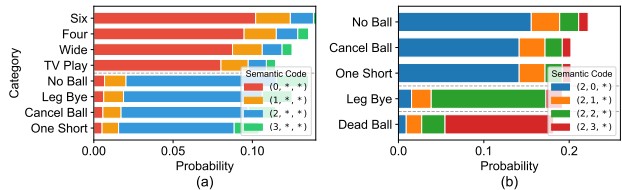

*Figure 4.* (a) The ground-truth category distribution for all the samples on the Cricket dataset, coloured by the value of the first code. (b) The category distributions for samples having the semantic code as $(2, *, *)$. The categories are colour-coded based on the second code.

temporal semantic codes and the length of the prediction trajectory. With a shallow hierarchy, the discretization captures only the coarsest temporal features, failing to resolve the fine-grained features required to distinguish complex classes. Increasing $L$ enables the model to recursively model the residuals, effectively establishing a coarse-to-fine prediction path that incorporates increasingly detailed structural information. As illustrated in Figure 2, the performance on ArticularyWordRecognition improves from $L = 1$ to $L = 2$, and then slightly degrades when further increasing the depth. A similar trend is observed on InsectWingbeat, where $L = 3$ attains the best accuracy. These observations indicate that when $L$ becomes excessive, deeper layers tend to encode high-frequency noise rather than semantic structure, making it harder for the LLM to predict without valid discriminatory signals. Empirically, setting $L = 2$–$3$ yields the most robust trade-off, enabling the model to maintain a compact prediction trajectory, without making the prediction process so complex that it overfits noise.

**Impact of Codebook Size $K$.** We further examine the size $K$ of the codebooks, varying $K \in \{16, 32, 64, 128, 256\}$. A severely constrained codebook imposes a substantial information bottleneck, forcing a wide range of distinct temporal patterns to be mapped onto a small set of prototypes, which leads to underfitting and poor class separability. As shown in Figure 3, on ArticularyWordRecognition, the accuracy is highest when $K = 16$ and gradually declines as $K$ increases, indicating that a compact yet expressive codebook is sufficient for this relatively simple dataset. On InsectWingbeat, however, a slightly larger codebook yields the best performance, suggesting that more complex and noisy dynamics benefit from a richer set of prototypes. In general, as $K$ increases, the latent space becomes more expressive, allowing a finer partition of the temporal embedding space and a more accurate approximation of the underlying continuous representations. However, beyond a moderate capacity, the marginal gains diminish, and we begin to observe signs of codebook sparsity, where a non-trivial fraction of prototypes are rarely utilised. This inefficiency reduces the model's capacity and can destabilise optimisation. Across datasets, we find that relatively small to medium codebook sizes (e.g., $K = 16$–$128$) provide a

favourable trade-off between representational richness and stable training.

### 5.5. Visualization Analysis

To verify that the learned temporal semantic codes capture meaningful hierarchical semantics, we visualise the code distributions for different action categories on the Cricket dataset, which consists of recordings of 12 different hand-signal gestures in the cricket game. At the first level (Figure 4(a)), we observe a clear semantic separation: two-handed actions are predominantly assigned to codes of the form $(0, *, *)$, whereas one-handed actions concentrate around $(2, *, *)$. This indicates that the first quantization stage effectively captures the coarse-grained semantic distinction between single-handed and two-handed actions. Within the subset of one-handed actions, the second-level code distribution (Figure 4(b)) further refines the semantics. Actions primarily involving upper-body movements cluster around $(2, 0, *)$, while actions engaging both the upper and lower body are mainly mapped to $(2, 2, *)$. This demonstrates that deeper quantization stages progressively encode finer-grained temporal semantics. Overall, these visualisations support our hypothesis that temporal semantic codes can serve as effective coarse-to-fine prediction trajectories for time series classification.

## 6. Limitations

Although our model achieves excellent performance on classification tasks, it has not yet been validated on other time-series tasks, such as long-term forecasting and anomaly detection. In addition, Hi-Time follows a three-stage training pipeline. While this design improves training stability, it prevents downstream prediction and classification losses from directly shaping the encoder and the quantizer, potentially leaving the learned semantic codes suboptimal for the end task. Therefore, in future work, we will explore the applicability of Hi-Time to a broader range of time-series tasks and investigate end-to-end improvements to the framework.

# 7. Conclusion

In this paper, we propose **Hi-Time**, a novel framework that equips Large Language Models (LLMs) with hierarchical latent prediction capabilities for multivariate time series classification. By shifting the chain-of-thought paradigm from explicit textual prompts to a learnable latent space of hierarchical semantic codes, our approach effectively overcomes the semantic ineffability inherent in temporal data. Specifically, we introduce a multi-view representation fusion that acquires robust representations by integrating multi-scale, cross-variable, and temporal dynamics information. To structure the prediction process, we discretise these representations into hierarchical temporal semantic codes via residual quantisation, establishing a ground-truth trajectory from coarse to fine granularity. Furthermore, we implement a hierarchical latent prediction strategy that enables the LLM to progressively predict these semantic codes to guide the final classification decision. Extensive experiments across 10 diverse benchmarks demonstrate that Hi-Time achieves state-of-the-art performance, validating the importance of structured hierarchical supervision in time-series classification.

# Acknowledgements

We thank the anonymous reviewers for their helpful feedback, and all the contributors of the original datasets. The work described in this paper was partially funded by the National Natural Science Foundation of China (Grant Nos. 62272173,62273109), the Natural Science Foundation of Guangdong Province (Grant Nos. 2024A1515010089, 2022A1515010179), the Science and Technology Planning Project of Guangdong Province (Grant No. 2023A0505050106), and the National Key R&D Program of China (Grant No. 2023YFA1011601).

# Impact Statement

Our proposed work, Hi-Time, aims to advance the field of Machine Learning by providing a more efficient model for multivariate time series classification across various applications. There are many potential societal consequences of our work, none of which we feel must be specifically highlighted here.

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
