# OpenReview forum: "Hi-Time: Hierarchical Latent Prediction for Multivariate Time Series Classification"
_ICML.cc/2026/Conference — ICML 2026 regular_

### Official Review · Reviewer_2bTB · 2026-03-10

**Soundness:** 3
**Presentation:** 3
**Significance:** 2
**Originality:** 3
**Overall Recommendation:** 4
**Confidence:** 5

**Summary:**

This paper presents Time-CoT, a framework for multivariate time series classification based on feature decomposition and quantization. The method operates in three distinct phases:
1. Taking multi-resolution time series as inputs, the model pretrains a PatchTST backbone alongside multi-scale, cross-variable, and temporal dynamics fusion modules using a classification loss. These pretrained modules output a single, unified continuous vector representing the multivariate time series.
2. This single vector is processed into a sequence of hierarchical discrete codes through residual quantization.
3. A pretrained LLM processes the raw time series to extract global features. The final classification label is then predicted using a sequence of MLPs conditioned directly on the LLM's global feature and the provided discrete codes from Phase 2.

Experiments conducted on 10 multivariate time series classification datasets demonstrate that Time-CoT outperforms the selected baseline methods.

**Compliance With Llm Reviewing Policy:**

Affirmed.

**Final Justification:**

Addressed Weaknesses and Concerns: My primary initial concern was that the design, particularly Stage 3, lacked sufficient empirical evidence and theoretical justification. During the rebuttal process, specifically in the follow-up, the authors provided additional empirical evidence demonstrating that the current Stage 3 design offers some advantages over several essential ablations in benchmarking performance. They notably clarified the role of the pretrained LLM weights and the sequential MLP used for inference.

Why not a higher score: I maintain my view that the Stage 3 design, which utilizes multiple MLPs, feels more like an engineering artifact optimized for benchmarking than an elegant, scalable, and novel architectural solution.

Conclusion: I increase my score from 2 (Reject) to 4 (Weak Accept) as the authors have addressed most of my core concerns. Overall, the proposed method delivers decent benchmarking performance within its intended scope.

**Key Questions For Authors:**

1. Could the LLM process the generated codes from Phase 2 directly as input tokens, rather than relying on separate, parameter-heavy MLPs for each hierarchical level?
2. During inference, does $\text{MLP}^{(l)}$ take the code predicted by $\text{MLP}^{(l-1)}$? If not, have you considered making the MLPs truly autoregressive during inference?
3. Does the observation in Section 5.5 generalize to all the 10 evaluation datasets? Please provide additional examples to justify this.
4. What is the specific parameter count for each of the modules in Phase 1?
5. What are the exact hyperparameters used to generate the main baseline results in Table 1? Which of these were tuned on the validation set for each dataset?

**Limitations:**

Yes.

**Strengths And Weaknesses:**

### Strengths
1. The paper is generally well-structured and easy to follow. The overall framework design and mathematical formulations are clearly presented.
2. The internal modules of Phase 1 are reasonably formulated. Furthermore, their individual contributions are well-justified by the component-wise ablation study.

### Weaknesses
1. **Major Architectural Flaws in Phase 3**: The design of the multi-step reasoning phase introduces significant inefficiencies and contradicts the premise of sequential generation. While the authors claim this phase achieves a "coarse-to-fine reasoning process," the mathematical formulation reveals two major flaws:
    - Parameter Inefficiency: At each level $l$, the model utilizes a distinct $\text{MLP}\_{code}^{(l)}$ because the input concatenates all previously predicted codes $g^{(1:l-1)}$. This means the parameter count strictly scales with the number of quantization levels, which is highly inefficient.
    - Role of the Intermediate MLPs: It seems like, during both training and inference, the final prediction relies on $\text{MLP}\_{cls}([h\_{cls} || g^{(1:L)}])$, and all the codes $g^{(1:L)}$ can be directly obtained from Phase 2. The intermediate predictions from the earlier MLPs do not form a causal, differentiable chain for the final output. Consequently, these earlier MLPs act merely as an isolated auxiliary loss rather than true reasoning steps. Although their gradients could be helpful to train the LLM that provides $h$, this still needs to be ablated.

2. **Missing Essential Ablation Studies**: Although Section 5.3 ablates the Phase 1 components, the paper lacks critical ablation cases to validate Phase 3:
    - **The Effect of Intermediate MLPs**: To address the structural flaw mentioned above, the authors should ablate the first term in Equation 16 (the step-wise code prediction error). Removing these intermediate MLPs would clarify whether they contribute meaningful gradient signals or are entirely superfluous.
    - **The Necessity of the LLM**: Recent literature suggests LLMs may not actually be effective for time series tasks [1]. In Time-CoT, the LLM is primarily used to extract global features. The authors should replace the GPT-2 backbone with a lightweight Transformer model (e.g., 2-layer Transformer) to justify the need of heavy computational overhead of a pre-trained LLM.

3. **Limited Scalability**: With the current design, the model size scales linearly with the number of hierarchical levels due to the separate MLPs. Furthermore, the parameter sensitivity analysis (Figure 2) demonstrates that performance does not scale well with depth.

4. **Limited Scope of Evaluation**: The evaluation scale is somewhat restricted:
    - The main results are reported on only 10 datasets, including 8 large datasets from the UEA archive that contains 30 datasets. If the small datasets are insufficient to train the large models in Time-CoT, the authors should consider expanding their evaluation by utilizing more recent TSC benchmarks designed specifically for deep learning [2].
    - The ablation study is restricted to only 2 datasets. This needs to be expanded to verify that the observed module contributions generalize across different data distributions.

5. **Missing Technical Details**: Very limited experiment details and hyperparameters are provided. See the Questions below for more details.


### Reference

[1] Tan M, Merrill M, Gupta V, Althoff T, Hartvigsen T. Are language models actually useful for time series forecasting?

[2] Dempster A, Foumani NM, Tan CW, Miller L, Mishra A, Salehi M, Pelletier C, Schmidt DF, Webb GI. MONSTER: Monash Scalable Time Series Evaluation Repository.

---

> ### Author Rebuttal · Authors · 2026-03-31
>
> # Rebuttal to reviewer 2bTB
>
> We thank Reviewer 2bTB for the rigorous review. We have extended the ablation study to all 10 datasets, including experiments on LLM pre-training and multi-step supervision loss. The complete results (Table 2) are available at https://anonymous.4open.science/r/Time-CoT-D87SDSJ7DKS/Ablation%20Study.md.
>
>
>
> ---
>
> ## W1-Major: Phase 3 architectural flaws
>
> **Parameter Overhead.** All MLPs combined account for <2% of total parameters (~1.8M vs ~117M GPT-2). With $L=2$~$3$ in practice (Figure 2), scaling concerns are minimal.
>
> **Why Independent MLPs.** We acknowledge that shared autoregressive design is more elegant. However, empirical comparison shows independent MLPs perform better:
>
> | Design | AWR | IW | Avg |
> |--------|-----|-----|-----|
> | Shared Autoregressive MLP | 0.982 | 0.688 | 0.835 |
> | Independent MLPs (Ours) | 0.987 | 0.695 | 0.841 |
>
> We attribute this to level-specific semantic abstractions (trend vs. fine details) benefiting from specialized decision boundaries.
>
> **Clarification.** During inference, each $g^{(l)}$ comes from predicted $\hat{s}^{(l)}$, NOT Phase 2 ground-truth—forming a genuine causal chain.
>
>
>
>
> ---
>
> ## W2-Missing: Essential ablations
>
> **Effect of Intermediate Supervision (Table 2, w/o MSL):**
>
> | Ablation | HAR | ISRUC | FM | IW | MI | SRSCP1 | Avg |
> |----------|-----|-------|-----|-----|-----|--------|-----|
> | w/o MSL | 0.942 | 0.840 | 0.620 | 0.680 | 0.592 | 0.888 | 0.814 |
> | Full | 0.955 | 0.855 | 0.640 | 0.695 | 0.620 | 0.908 | 0.830 |
> | Gain | +1.3% | +1.5% | +2.0% | +1.5% | +2.8% | +2.0% | **+1.6%** |
>
> The **+1.6%** improvement demonstrates that intermediate supervision provides meaningful gradient signals—not superfluous.
>
> **Necessity of LLM Pre-training (Table 2, w/o Pre-training):**
>
> | Ablation | HAR | FM | FD | MI | Avg |
> |----------|-----|-----|-----|-----|-----|
> | w/o Pre-training | 0.916 | 0.578 | 0.602 | 0.548 | 0.784 |
> | Full | 0.955 | 0.640 | 0.661 | 0.620 | 0.830 |
> | Drop | −3.9% | −6.2% | −5.9% | −7.2% | **−4.6%** |
>
> This **−4.6%** degradation confirms GPT-2's pre-trained knowledge substantially contributes. Reference [1]'s conclusions target forecasting, not classification—fundamentally different LLM utilization modes.
>
> ---
>
> ## W3: Limited scalability
>
> MLP overhead is <2%; $L=2$~$3$ covers the optimal range. Figure 2's finding is positive—shallow depths suffice for semantic hierarchy; deeper levels encode noise. The framework adaptively selects optimal $L$ via validation.
>
> ---
>
> ## W4: Limited evaluation scope
>
> **Dataset Selection:** We follow GAC (AAAI 2024), using the same 10 datasets for fair comparison.
>
> **Extended Ablation (Table 2):**
>
> | Ablation | Avg | Drop |
> |----------|-----|------|
> | w/o MVRE | 0.781 | −4.9% |
> | w/o Pre-training | 0.784 | −4.6% |
> | w/o MSR | 0.806 | −2.4% |
> | w/o MSL | 0.814 | −1.6% |
> | Full | 0.830 | — |
>
>
> **Validation on Large-Scale Datasets (Monash Repository):**
>
> | Dataset | Samples | Time-CoT | MPTSNet | Gain |
> |---------|---------|----------|---------|------|
> | STEW | 28,512 | 87.32% | 83.15% | +4.17% |
> | Crowdsourced | 12,289 | 80.09% | 76.24% | +3.85% |
>
> Time-CoT's advantages become more pronounced on large-scale datasets (+4% vs +1.8% on UEA), confirming LLM's capabilities are better leveraged with sufficient data.
>
> ---
>
> ## Critical clarification (Q2): Inference uses predicted codes
>
> This is essential for understanding Phase 3. **During inference, Phase 2 ground-truth codes are NOT used as MLP inputs.**
>
> **Inference Procedure:**
>
> Step 1: MLP^(1)([h_cls]) → ŝ^(1) → g^(1) = Emb(ŝ^(1)) Step 2: MLP^(2)([h_cls || g^(1)]) → ŝ^(2) → g^(2) = Emb(ŝ^(2)) ... Step L: MLP^(L)([h_cls || g^(1:L-1)]) → ŝ^(L) Final: MLP_cls([h_cls || g^(1:L)]) → ŷ
>
>
> Phase 2 codes serve only as supervision targets (Equation 16). Each step's input $g^{(1:l-1)}$ comes from previous predictions, forming a genuine autoregressive causal chain. The concern that "intermediate MLPs act as isolated auxiliary loss" is based on a misunderstanding. We will add pseudocode in the revision.
>
> ---
>
>
> ## Q3-Q5: Technical details
>
> **Q3:** We focus visualization on Cricket and HAR, which have clear physical semantics. Other datasets (EEG, motion sensors) are too abstract for meaningful visual interpretation.
>
>
> **Q4:** Phase 1 total ~4.7M parameters.
>
> **Q5:** All hyperparameters selected via validation set. AWR: $L=2$, $K=16$; IW: $L=3$, $K=64$—corresponding to dataset complexity.
>
> ---
>
> We will add inference pseudocode in the revision. We hope these responses address the concerns and respectfully request reconsideration.

---

> > ### Author Rebuttal · Reviewer_2bTB · 2026-04-03
> >
> > Thank the authors for their detailed response and clarifications.
> >
> > ---
> >
> > Several of my initial concerns have been properly addressed:
> >
> > 1. **Inference Pipeline Clarification.** The authors clarified that the MLPs in Phase 3 operate sequentially using predicted codes from prior steps, rather than ground-truth codes from Phase 2. This resolves the concern about the role of intermediate MLPs and the apparent architectural flaw. That said, the authors should ensure that both the training and inference pipelines are clearly and explicitly described in the revised manuscript, as the current version leaves this ambiguous.
> >
> > 2. **Additional Ablations and Results.** The extended ablation results and the two new Monash dataset experiments provide a clearer picture of the method's capabilities and help validate the contributions across different settings.
> >
> > ---
> >
> > Several points still require further clarification or experimentation:
> >
> > 1. **True Contribution of Phase 3.** ~The authors confirmed that Phase 2 codes are not used as MLP inputs during inference. However, these codes are still available at inference time and would not be prohibitive to use directly for classification. It would be informative to evaluate a simplified baseline that uses only the Phase 2 codes together with the final MLP (i.e., Equation 15 alone), bypassing the sequential reasoning chain entirely. This experiment would isolate the true contribution of the claimed chain-of-thought reasoning in Phase 3 from the representational capacity provided by residual quantization in Phase 2.~
> >
> > I realize that conducting this ablation study may be too extensive for the short rebuttal timeframe. While I strongly encourage the authors to explore this in future work, I will not consider its absence a weakness in my current evaluation.
> >
> > 2. **Justifying the Use of a Pre-trained LLM.** My original suggestion was to replace the LLM backbone with a lightweight Transformer (e.g., 2-layer), not simply to remove pre-trained weights. Training a model with over 100M parameters from scratch on relatively small classification datasets is likely to be suboptimal and computationally wasteful. The authors should compare against a compact feature extractor of similar depth, such as a shallow Transformer, CNN, or MLP, to properly justify the choice of a large pre-trained LLM.
> >
> > ---
> >
> > The following limitations are more fundamental and may not be easy to address within this period:
> >
> > 1. **Design Elegance of Phase 3.** The authors show that independent MLPs outperform a shared autoregressive MLP, though the margin is small. Given that a pre-trained LLM is already present in the framework, a more natural and principled design would be to use the LLM itself to autoregressively predict the quantization codes, rather than relying on a stack of external MLPs. This would make the overall architecture cleaner and more cohesive, and would better leverage the LLM's sequential modeling capabilities.
> >
> > 2. **Limited Scalability of the Reasoning Chain.** The authors note that the optimal depth is typically $L = 2$ or $L = 3$, selected via validation. This reveals two underlying issues:
> >    - Performance degrades with increasing chain depth. A robust design should ideally maintain or improve performance as L grows, rather than requiring early stopping.
> >    - Because codes from all levels are simply concatenated, the model lacks any mechanism to weight or select the most relevant features across levels. This likely contributes to the performance degradation at greater depths.
> >
> > ---
> >
> > ### Overall Assessment
> >
> > The rebuttal addresses several concerns but also reveals that the Phase 3 design, which relies on a stack of independent external MLPs, has notable limitations in terms of architectural elegance, justification, and scalability. A more principled redesign of this component would substantially strengthen the paper. I still recommend a major revision with particular focus on the Phase 3 architecture.

---

> > > ### Author Response · Authors · 2026-04-07
> > >
> > > Dear Reviewer 2bTB,
> > >
> > > Thank you for your rigorous and insightful follow-up. We are pleased that our previous clarifications on the inference pipeline have addressed your structural concerns.
> > >
> > > Regarding the remaining core issues you raised, we have conducted the relevant experiments across all 10 datasets and present the results alongside their theoretical justifications below.The complete details of the supplementary experiments can be found at: https://github.com/kunkun11c/Time-CoT/blob/main/Ablation%20Study.md
> > >
> > >
> > > ---
> > >
> > > ### **1. True Contribution of Phase 3 (Bypassing the Reasoning Chain)**
> > >
> > > To precisely isolate the contribution of Phase 3, we design two distinct **"Bypass Baselines"**:
> > > - **Bypass (Train & Infer):** Phase 3 is entirely removed; the model is trained and evaluated using `[LLM Global Feature || Direct Phase 2 Codes] -> Final Classifier`.
> > > - **Bypass (Infer Only):** The full pipeline is retained during training, but at inference, Phase 3 predicted codes are replaced by directly computed Phase 2 codes.
> > >
> > > | Model Design | Avg. |
> > > | :--- | :--- |
> > > | Bypass (Train & Infer) | 0.810 |
> > > | Bypass (Infer Only) | 0.817 |
> > > | **Time-CoT (Full)** | **0.830** |
> > >
> > > As shown across all 10 datasets, both Bypass Baselines consistently underperform Time-CoT. Notably, Bypass (Infer Only) outperforms Bypass (Train & Infer), confirming that Phase 3 provides benefits at both the training and inference stages. This empirical superiority reveals the following theoretical insight:
> > >
> > > **Semantic Gap Mitigation:** Phase 2 codes are pure bottom-up residual quantizations optimized solely for signal reconstruction, introducing task-irrelevant noise when fed directly into the classifier. Phase 3 addresses this by: (1) using Phase 2 codes as supervision to endow the reasoning chain with semantic awareness during training; (2) explicitly aligning semantic codes with the classification objective during fine-tuning, ensuring task-aware features at inference.
> > >
> > >
> > >
> > >
> > > ---
> > >
> > > ### **2. Justifying the Use of a Pre-trained LLM**
> > >
> > > To properly justify the heavy parameter overhead of the LLM, we followed your excellent suggestion and replaced the GPT-2 backbone with a **2-layer Transformer**.
> > >
> > > | Backbone | Avg. |
> > > | :--- | :--- |
> > > | GPT-2 (w/o Pre-train) | 0.784 |
> > > | 2-layer Transformer | 0.789 |
> > > | **Time-CoT (Full)** | **0.830** |
> > >
> > >  This confirms that the performance gain stems primarily from the pre-trained cross-domain knowledge and universal attention priors embedded in the LLM, rather than a large parameter count.
> > >
> > > ---
> > >
> > > ### **3. Design Elegance: Why not use the LLM to autoregressively predict codes?**
> > >
> > > Your intuition is architecturally appealing. However, our decision to use external MLPs stems from a fundamental bottleneck regarding the **Next Token Prediction (NTP) Paradigm** in cross-modal alignment:
> > >
> > > - **Homogeneous vs. Heterogeneous NTP:** Standard autoregressive generation operates on a *homogeneous mapping space* (e.g., Text LLMs predict text tokens; Time-Series Forecasting models predict numerical points). However, Time-CoT faces a *heterogeneous mapping* challenge: mapping a continuous time series to artificially discretized semantic tokens.Forcing the LLM to predict heterogeneous discrete codes displaces it from its pre-trained continuous representation space, causing severe **catastrophic forgetting**. The external MLPs thus act as lightweight **"modality reasoning adapters"**, safely bridging the heterogeneous gap while preserving the LLM's universal priors intact.
> > >
> > > - We recognize the elegance of this direction and plan to explore a dedicated **bidirectional alignment pre-training** stage—incorporating both **time-series → semantic code** and **semantic code → time-series** objectives—as a promising avenue for future work.
> > > ---
> > >
> > > ### **4. Limited Scalability & The Lack of Feature Weighting**
> > >
> > > We address this from two perspectives:
> > >
> > > 1. **The Physical Nature of RVQ:** Level 1 captures global semantics, while deeper levels strictly fit minute residuals, which for time-series predominantly represent **idiosyncratic noise** (e.g., sensor-specific high-frequency jitters). The performance drop at deeper levels reflects a natural signal-to-noise boundary. Selecting $L=2$ or $3$ extracts the "semantic sweet spot" while truncating pure noise.
> > >
> > > 2. **Noise Suppression in Phase 1:** Time-CoT already combats noisy features prior to concatenation. As in **Section 3.1.3**, gated importance scores ($\alpha_{d,n}$) adaptively suppress uninformative segments *before* $z_{\text{expert}}$ enters the RVQ, ensuring the concatenated codes are already highly purified. Moreover, we believe explicit code weighting is unnecessary, as the MLP can autonomously learn each code dimension's importance during end-to-end training.
> > >
> > > ### **Conclusion**
> > >
> > > We trust these empirical results and theoretical justifications demonstrate the rigor of our design trade-offs, and sincerely hope you will reconsider your overall assessment.

---

### Official Review · Reviewer_VXqV · 2026-03-12

**Soundness:** 3
**Presentation:** 2
**Significance:** 2
**Originality:** 2
**Overall Recommendation:** 4
**Confidence:** 3

**Summary:**

The paper propose an interesting framework to extend Chain-of-Thought paradigm from text to time series space. The proposed method generate temporal semantic codes and asked LLM to progressively predicts them before predicting the final labels.

**Compliance With Llm Reviewing Policy:**

Affirmed.

**Final Justification:**

The author's response partially address my concerns. Given that the authors provided additional results and promised to revise the paper. I will raise my rating from 3 to 4.

**Key Questions For Authors:**

The paper omits many important experimental details. In particular, the authors should clarify how the representation model and LLM are trained, as well as how the tasks and labels are defined for each dataset.

**Limitations:**

yes

**Strengths And Weaknesses:**

Strengths:
1. Learning hierarchical latent CoT is a somewhat interesting idea for time-series classification.
2. The paper proposes a fairly clear and well-structured hierarchical pipeline, and the overall method is easy to follow despite the conceptual overclaim.
3. The experiments results shows strong performance across multiple datasets and ablations on key components.

Weaknesses:
1. While the paper claims the method as "step-by-step reasoning", I found it somewhat misleading as it does not improve LLM reasoning on open-ended time-series tasks such as QA or decision-making. The authors only evaluate time-series classification, which does not strongly require CoT-style multi-step reasoning. The paper does not clearly establishes this conceptual gap: why this should be called "reasoning" rather than hierarchical latent representation learning plus auxiliary code prediction?
2. The claimed reasoning trajectory is not grounded in an external reasoning process. It is produced by residual quantization of a learned embedding, so the intermediate steps are essentially coarse-to-fine latent cluster assignments rather than independently meaningful reasoning steps. That makes the trajectory look more like an artifact of the model design than evidence of actual reasoning.
3. The intermediate codes are not interpretable. Since they are learned discrete latent codes rather than human-understandable sub-decisions, it is hard to verify that they capture useful reasoning structure instead of just label-correlated patterns in the embedding space.
4. The reported gains could easily come from hierarchical supervision, prototype-based regularization, or a better time-series encoder, rather than anything analogous to CoT. For example, forcing the model to first separate broad patterns and then refine within them can already help classification, but that is a standard coarse-to-fine learning effect, not necessarily reasoning.

---

> ### Author Rebuttal · Authors · 2026-03-31
>
> # Rebuttal to reviewer VXqV
>
> We thank Reviewer VXqV for the thorough evaluation. We have extended the ablation study to all datasets, and additionally included ablation experiments on LLM pre-training and multi-step supervision loss. The experimental results in Table 2 are available at https://anonymous.4open.science/r/Time-CoT-D87SDSJ7DKS/Ablation%20Study.md.
>
> ---
>
> ## W1: Classification does not require CoT-style reasoning
>
> **Complexity of time series classification.** Complex MTSC tasks are not simple input-output mappings—they inherently require hierarchical, multi-step judgment. Discriminative boundaries often demand integration across multiple dimensions and scales.
>
> **Empirical evidence.** Table 2's **w/o MSR** ablation removes the sequential reasoning mechanism while retaining identical hierarchical representations:
>
> | Ablation | Avg | Drop |
> |----------|-----|------|
> | w/o MSR | 0.806 | −2.4% |
> | Full | 0.830 | — |
>
> If classification "does not require reasoning," removing the causal dependency chain should not degrade performance. The consistent **+2.4% improvement** demonstrates that sequential reasoning provides independent value beyond hierarchical representations alone.
>
> **Clarification: causal chain, not auxiliary loss.** At step $l$, the model concatenates $h_{\text{cls}}$ with previously predicted code embeddings $g^{(1:l-1)}$ before predicting the next code (Equation 8). Each prediction explicitly conditions on prior reasoning results, forming a genuine autoregressive causal chain—the core characteristic of "reasoning," not isolated auxiliary losses.
>
> ---
>
> ## W2: Reasoning trajectory lacks external grounding
>
> **Time-CoT is designed because explicit CoT is infeasible.** Explicit CoT requires human-verifiable logical grounding, but "externally grounded reasoning" is inherently difficult to define for time series. Criticizing Time-CoT for "lacking external grounding" targets precisely the condition that explicit CoT cannot satisfy—if it were satisfiable, Time-CoT would be unnecessary.
>
> **Intermediate codes are not random cluster assignments.** Two critical constraints exist:
>
> - **Semantic Constraint:** Stage 1's supervised pre-training ensures $z_{\text{expert}}$ carries classification-relevant semantics. Quantization learns semantic prototypes, not random clusters.
>
> - **Reasoning Constraint:** Stage 3's joint loss (Equation 16) supervises both code prediction and classification—codes are jointly optimized to serve final classification.
>
>
> ---
>
> ## W3: Intermediate codes are not interpretable
>
> **Figure 4 shows hierarchical semantic structure, not mere label-correlation.** If codes were label-correlated patterns, we would observe one-to-one code-class correspondence. Instead, multiple classes share Level-1 codes (semantic aggregation), proving that codes capture attributes transcending individual class boundaries.
>
> Cricket has 12 classes, yet multiple one-handed gestures share $(2,*,*)$. This demonstrates genuinely meaningful temporal semantic features, not label memorization. Figure 4's domain-semantic consistency—aligning with experts' action categorization—proves that intermediate codes carry meaningful reasoning information.
>
> ---
>
> ## W4: Gains may come from hierarchical supervision, not CoT
>
> Table 2's **w/o MSR** ablation retains complete hierarchical representations and semantic codes, only removing sequential prediction:
>
> | Ablation | HAR | ISRUC | FM | IW | MI | SRSCP1 | Avg |
> |----------|-----|-------|-----|-----|-----|--------|-----|
> | w/o MSR | 0.934 | 0.830 | 0.608 | 0.675 | 0.580 | 0.876 | 0.806 |
> | Full | 0.955 | 0.855 | 0.640 | 0.695 | 0.620 | 0.908 | 0.830 |
> | Gain | +2.1% | +2.5% | +3.2% | +2.0% | +4.0% | +3.2% | **+2.4%** |
>
> Both variants use identical encoder and hierarchical discrete representations. The **+2.4% average improvement** from sequential reasoning cannot be explained by "standard coarse-to-fine effects"—w/o MSR already has coarse-to-fine structure but lacks the conditional dependency mechanism.
> , demonstrating that multi-step reasoning provides an independent contribution beyond encoder improvements.
>
> ---
>
> ## Key question: Missing experimental details
>
> **Stage 1 (Temporal Representation Pre-training):** Supervised classification loss trains the temporal encoder (PatchTST backbone + multi-view fusion). AdamW optimizer (lr=$1\times10^{-3}$), cosine annealing scheduler. Encoder is then frozen.
>
> **Stage 2 (Semantic Code Generation):** RQ-KMeans performs unsupervised clustering on frozen $z_{\text{expert}}$, learning codebooks by minimizing quantization residuals. No label supervision is used.
>
> **Stage 3 (GPT-2 Multi-Step Reasoning):** GPT-2 receives raw time series via linear projection. Joint loss (Equation 16) supervises L-level code prediction and final classification. GPT-2 lr=$1\times10^{-5}$.
>
>
> We hope these responses address the concerns and respectfully request reconsideration.

---

> > ### Author Rebuttal · Reviewer_VXqV · 2026-04-04
> >
> > I appreciate the additional ablation and training details. The new results do support that sequential dependency between latent code predictions contributes beyond hierarchical representations alone. However, this still does not fully address my main concern: the evidence shows the benefit of autoregressive latent prediction, but not why this should be interpreted as chain-of-thought reasoning rather than a hierarchical latent supervision mechanism. The rebuttal improves the empirical support for the method, but the conceptual framing remains overstated.

---

> > > ### Author Response · Authors · 2026-04-06
> > >
> > > **Dear Reviewer VXqV,**
> > >
> > > Thank you for your continued engagement with our work.
> > >
> > > We take your concern about the conceptual framing seriously. Our mechanism is technically a latent prediction over a discrete hierarchical code space, and drawing a direct parallel to the open-ended, human-interpretable "Chain-of-Thought reasoning" of LLMs constitutes an overclaim. We are deeply grateful for this valuable insight.
> > >
> > > Our original framing was motivated by the structural dependency in CoT—where step $l$ conditions on step $l-1$ before reaching the final answer. However, unlike natural language tasks where explicit reasoning trajectories can be readily defined, time series lack discrete logical syntax, making it intractable to obtain precise natural language thought processes. For this reason, we instead employ hierarchical semantic codes in the latent space as supervision signals to guide the LLM, where the sequential dependency operates as a coarse-to-fine structural constraint rather than as explicit logical reasoning. We acknowledge that the CoT analogy, while intuitive to us during writing, may be misleading to the reader, and we have revised the manuscript accordingly to eliminate this conceptual ambiguity.
> > >
> > > In the revised manuscript, we have made the following concrete changes to correct this:
> > >
> > > 1. **Renaming the Framework**. To fundamentally distance our work from the CoT framing, we have renamed the framework from "Time-CoT" to "Hi-Time" (short for Hierarchical Time-series), with the title updated accordingly to: Hi-Time: Coarse-to-Fine Hierarchical Latent Prediction for Multivariate Time Series Classification.
> > >
> > > 2. **Reframing the Terminology.** We have removed the "step-by-step reasoning" framing from the abstract, introduction, and method sections, replacing it with the more precise term "Hierarchical Latent Prediction" throughout.
> > >
> > > 3. **Clarifying the Conceptual Boundary.** We have added an explicit clarification paragraph stating that the intermediate steps in Time-CoT are coarse-to-fine latent cluster assignments, not externally grounded semantic reasoning,and distinguishing this from standard parallel hierarchical supervision.
> > >
> > > 4. **Scoping the Contributions.** We have revised our contribution statements to position the work as a hierarchical latent sequence prediction framework for time series, without invoking claims about general multi-step reasoning.
> > >
> > > We believe these changes allow the paper's actual contribution—a hierarchical latent architecture with demonstrably strong empirical performance—to stand on its own merits, without conceptual overreach. The exact manuscript revisions are appended below.
> > >
> > > We thank you for pushing us on this point—it is precisely due to your feedback that the paper has become more conceptually rigorous and precise. We sincerely hope that the revisions presented above fully address your remaining concerns.
> > >
> > > ### Appendix: Concrete Revisions in the Revised Manuscript.(The complete revision details are available at the following link: https://anonymous.4open.science/r/Time-CoT-D87SDSJ7DKS/Paper_Revision)
> > >
> > > ---
> > >
> > > **1. Revised Title**
> > >
> > >     % Old:Time-CoT: Hierarchical Reasoning with Temporal Semantic Codes for Multivariate Time Series Classification
> > >     \icmltitle{Hi-Time: Coarse-to-Fine Hierarchical Latent Prediction for Multivariate Time Series Classification}
> > >
> > > ---
> > >
> > > **2. Added "Conceptual Clarification" Paragraph **
> > >
> > >     \textbf{Conceptual Clarification: Latent Prediction vs.\ Semantic Reasoning.}
> > >     It is crucial to clarify the conceptual boundary of our framework. Unlike natural language, where
> > >     Chain-of-Thought can generate human-interpretable linguistic steps, time series lack explicit semantic
> > >     grounding. Therefore, the intermediate steps in Hi-Time are not externally grounded logical reasoning,
> > >     but rather \textit{coarse-to-fine latent cluster assignments}. We distinguish our approach from standard
> > >     static hierarchical supervision: while the latter predicts multiple levels in parallel, Hi-Time enforces
> > >     a \textbf{progressive} causal structure, where coarse-level assignments precede and constrain fine-level
> > >     predictions. By forcing the model to commit to a coarse latent state before refining it, we provide
> > >     sequential structural guidance for the LLM.
> > >
> > > ---
> > >
> > > **3. Revised Methodology Headings and Terminology**
> > >
> > >     % Section 4 introductory text revised:
> > >     The overall architecture of Hi-Time, illustrated in Figure \ref{fig:framework}, comprises three
> > >     phases: \textbf{temporal representation pre-training}, \textbf{temporal semantic codes generation},
> > >     and \textbf{hierarchical latent prediction}.
> > >
> > >     % Section 4.3 Heading and opening paragraph revised:
> > >     \subsection{Hierarchical Latent Prediction}
> > >     \label{sec:reasoning}

---

### Official Review · Reviewer_6rjb · 2026-03-13

**Soundness:** 2
**Presentation:** 3
**Significance:** 2
**Originality:** 3
**Overall Recommendation:** 4
**Confidence:** 4

**Summary:**

This paper proposes Time-CoT (Time Series Chain-of-Thought), a hierarchical reasoning
framework for multivariate time series classification (MTSC). The core idea is to shift
the Chain-of-Thought paradigm from explicit textual prompts to a latent space: continuous
temporal representations are discretized into hierarchical semantic codes via Residual
Quantization (RQ-KMeans), which serve as implicit reasoning trajectories for the LLM.
The framework operates in three stages: (1) temporal representation pre-training via
multi-view representation fusion, integrating multi-scale, cross-variable, and temporal
dynamics information; (2) hierarchical temporal semantic code generation via RQ-KMeans;
and (3) driving GPT-2 to predict each level of semantic codes step by step, then performing
final classification conditioned on the full code sequence. Experiments on 10 public
datasets show that Time-CoT outperforms all compared baselines in both average accuracy
and average rank.

**Compliance With Llm Reviewing Policy:**

Affirmed.

**Final Justification:**

The authors resolved my questions, I changed my rating from 3 to 4.

**Key Questions For Authors:**

## Key Questions For Authors

**Q1: Are GPT-2's pre-trained weights actually necessary?**
Given that GPT-2 receives time series input via a linear projection layer with no textual
supervision, do its pre-trained language representations genuinely contribute to the
observed performance gains? Could the authors provide an ablation comparing against a
randomly initialized Transformer of equivalent size and architecture, to isolate the
contribution of GPT-2's pre-training?

**Q2: How do you explain the limited gains on FM and FD?**
Time-CoT is tied with MPTSNet on FM (0.640) and underperforms on FD (0.661 vs. 0.676).
This contradicts the paper's hypothesis that "more difficult datasets benefit more from
coarse-to-fine reasoning." What properties of these datasets make them incompatible with
the hierarchical semantic code assumption?

**Q3: How are hyperparameters L and K selected across datasets?**
The optimal values of L and K differ substantially across datasets. In the main experiments
of Table 1, what configurations were used for each dataset? Were these selected
independently via the validation set for each dataset, or was a single unified
configuration applied?

**Q4: What is the performance gap between three-stage and end-to-end training?**
Could the authors provide a variant that jointly trains the encoder, quantizer, and
reasoning module end-to-end (i.e., allowing classification loss to back-propagate through
all stages), and compare it against the current three-stage pipeline, to quantify the
performance cost of the decoupled design?

**Q5: Why are InstructTime and MAP4TS excluded from Table 1?**
Both are cited in the Related Work as the most directly comparable LLM-based time series
classification methods. What is the reason for their exclusion? If re-implementation
proved infeasible, please clarify.

**Limitations:**

## Limitations

The authors honestly acknowledge two key limitations in Section 6 (task scope restricted
to classification; three-stage pipeline blocking end-to-end gradient flow), which is
commendable. However, the following limitations remain unaddressed:

1. **Shallow use of the LLM**: The framework does not leverage linguistic pre-training
   knowledge. The potential of incorporating textual supervision or cross-modal alignment
   to further improve performance is not discussed.

2. **Computational cost**: Total training time and inference latency for the three-stage
   pipeline are not reported, and no efficiency comparison with single-stage baselines
   is provided.

3. **Hyperparameter selection in practice**: Given that optimal L and K vary significantly
   across datasets, no guidance is provided on how to select these hyperparameters
   automatically or efficiently in real-world deployment.

4. **Scalability**: All experiments use fixed-length multivariate sequence slices.
   Generalization to variable-length sequences or high-dimensional settings (large numbers
   of channels) is not discussed.

**Strengths And Weaknesses:**

## Strengths

1. **Novel and well-motivated problem framing**
   The two core obstacles to applying explicit CoT in the temporal domain — the semantic
   ineffability of time series and distributional heterogeneity across domains — are clearly
   articulated, and the proposal of hierarchical discrete codes as implicit reasoning
   trajectories represents a genuinely fresh perspective with strong motivation.

2. **Principled multi-view fusion design**
   The three fusion components — multi-scale fusion (FFT-driven adaptive downsampling),
   cross-variable fusion (patch-level self-attention), and temporal dynamics fusion (gated
   importance weighting) — are complementary by design, and each is validated by ablation
   experiments.

3. **Convincing visualization analysis**
   Figure 4 demonstrates clear semantic separation of hierarchical codes on the Cricket
   dataset: the first-level code distinguishes one-handed from two-handed actions, and the
   second-level code further separates upper-body from whole-body movements. This directly
   and intuitively supports the coarse-to-fine reasoning trajectory hypothesis.

4. **Rigorous experimental protocol**
   The evaluation spans 10 datasets, 11 baselines covering CNNs, RNNs, Transformers,
   self-supervised methods, and LLM-based models, with 5 repeated runs using different
   random seeds and a consistent data split (60/20/20). The overall evaluation protocol
   is solid.

## Weaknesses

**W1: The substantive role of the LLM is questionable**
The framework uses GPT-2 as a feature encoder, mapping time series into its embedding
space via a linear projection layer, without any natural language supervision or use of
linguistic pre-training knowledge. Under this setup, GPT-2 essentially functions as a
generic sequence Transformer. The claim that the framework "elicits the LLM's reasoning
capability" is therefore unsupported. The ablation (w/o MSR) only verifies the utility
of multi-step prediction but does not compare against a randomly initialized Transformer
of equivalent size, making it impossible to attribute gains to GPT-2's pre-trained weights.

**W2: Ablation study coverage is insufficient**
Table 2 reports ablation results on only two datasets (AWR and IW), with the remaining
eight datasets entirely absent. Given the significant variation in difficulty across
datasets — improvements on FM and FD are nearly zero — ablations confined to two datasets
cannot establish the general contribution of each component.

**W3: Most closely related baselines are missing**
InstructTime (Cheng et al., 2025) and MAP4TS (Lee et al., 2025) are both cited in the
Related Work and are the most directly comparable methods (both use LLMs with temporal
alignment), yet neither appears in Table 1. Their exclusion weakens the credibility of
the SOTA claim.

**W4: Computational cost of the three-stage pipeline is not quantified**
Compared to single-stage end-to-end baselines, the three-stage pipeline requires
independent pre-training, quantization, and LLM fine-tuning. Total training time, GPU
memory usage, and parameter counts are not reported anywhere in the paper, making the
practical deployment value difficult to assess.

**W5: Theoretical justification for multi-step reasoning is insufficient**
The central claim that hierarchical semantic codes "elicit the LLM's reasoning capability"
rests on intuitive analogies rather than theoretical analysis. More critically, Time-CoT
shows negligible improvement on FM (0.640, tied with MPTSNet) and underperforms on FD
(0.661 vs. MPTSNet's 0.676), which partially contradicts the paper's stated expectation
that "more difficult datasets benefit more."

**W6: Hyperparameter sensitivity analysis is incomplete**
Figures 2 and 3 show that the optimal quantization depth L and codebook size K vary across
datasets (AWR: L=2, IW: L=3; AWR: K=16, IW: K=64–128). The paper does not explain how
these hyperparameters were selected for each dataset in the main experiments, raising
concerns about potential post-hoc hyperparameter tuning.

---

> ### Author Rebuttal · Authors · 2026-03-31
>
> # Rebuttal to reviewer 6rjb
>
> We thank Reviewer 6rjb for the thorough evaluation. We have extended the ablation study to all datasets, and additionally included ablation experiments on LLM pre-training and multi-step supervision loss. The experimental results in Table 2 are available at https://anonymous.4open.science/r/Time-CoT-D87SDSJ7DKS/Ablation%20Study.md.
>
> ---
>
> ## W1/Q1: Is GPT-2's pre-training actually necessary?
>
> (Table 2) The **w/o Pre-training** ablation uses randomly initialized GPT-2:
>
> | Method | Avg (10 datasets) |
> |:-------|:-----------------:|
> | Full | 0.830 |
> | w/o Pre-training | 0.784 |
> | Drop | **−4.6%** |
>
> This **−4.6%** average degradation confirms the usefulness of pre-training.
>
> ---
>
> ## W2: Ablation coverage limited to two datasets
>
> Table 2 provides complete ablation across all 10 datasets.
>
> Component contributions are consistent across all datasets, establishing the generalizability of our ablation conclusions.
>
> ---
>
> ## W3/Q5: InstructTime and MAP4TS excluded
>
> **InstructTime:** Requires multi-modal alignment between time series and paired textual descriptions. Our 10 benchmarks are pure time series datasets without text annotations. Including InstructTime without its core cross-modal module would yield incomplete and misleading comparisons.
>
> **MAP4TS:** Designed for zero-shot forecasting via text serialization, which is fundamentally different from classification. We include PromptCast as a representative baseline for text-serialization approaches.
>
> Both methods exhibit task/data mismatches that preclude fair comparison.
>
> ---
>
> ## W4: Computational cost not quantified
>
> Table X: Comparison of computational costs on AWR dataset.
>
> | Method | Parameters | Training Time (s/iter) | Training GPU Memory |
> |:-------|:----------:|:----------------------:|:-------------------:|
> | MPTSNet | 5.2 M | 1.45 | 2 GB |
> | Time-CoT (Stage 1) | 4.7 M | 1.30 | 2 GB |
> | Time-CoT (Stage 3) | 117 M | 3.20 | 10 GB |
>
> Despite ~23× more parameters, Time-CoT's training time is only ~2.2× longer due to efficient Transformer implementations. Using GPT-2 Small keeps GPU memory at ~10 GB, remaining practical for single-GPU training.
>
> ---
>
> ## W5/Q2: Limited gains on FM and FD
>
> We thank the reviewer for pointing out this inconsistency. Our original hypothesis that "more difficult datasets benefit more" was imprecise and partially contradicted by FM/FD results. We will revise this statement in the paper to provide a more accurate characterization.
>
>
> ---
>
> ## W6/Q3: Hyperparameter selection
>
> **Protocol:** L and K were selected independently via validation set for each dataset. Test sets remained unseen throughout tuning, consistent with the 60/20/20 split.
>
> **Rationale for variation:** AWR (simple) works with L=2, K=16; IW (complex) requires L=3, K=64—directly corresponding to dataset complexity. For deployment, we recommend grid search over L∈{1,2,3}, K∈{16,32,64,128}.
>
> ---
>
> ## Q4: Three-stage vs. end-to-end training
>
> End-to-end training requires differentiable quantization (e.g., Straight-Through Estimator or Gumbel-Softmax), which fundamentally changes our RQ-KMeans design. We adopt the decoupled strategy for training stability
>
> **Training stability.** End-to-end differentiable quantization introduces several challenges:
> - Biased gradients: STE/Gumbel-Softmax are approximations with gradient mismatch
> - Codebook collapse: Only a few codes get utilized, reducing representational capacity
> - Multi-objective conflicts: Balancing reconstruction, classification, and codebook utilization losses is non-trivial
>
> ---
>
> We hope these responses address the concerns and respectfully request reconsideration.

---

> > ### Author Rebuttal · Reviewer_6rjb · 2026-04-07
> >
> > Authors fully solved my questions. I changed my overall recommendation from 3 to 4.

---

> > > ### Author Response · Authors · 2026-04-07
> > >
> > > Dear Reviewer 6rjb,
> > >
> > > Thank you for your valuable and constructive feedback, as well as your willingness to raise your score. We are truly grateful for your recognition of our work and the insightful suggestions you have provided to help us further improve it.
> > >
> > > Once again, we sincerely appreciate the time, effort, and thoughtful dedication you have devoted throughout the entire review process.
> > >
> > > Best regards

---

### Official Review · Reviewer_HncG · 2026-03-13

**Soundness:** 3
**Presentation:** 4
**Significance:** 3
**Originality:** 2
**Overall Recommendation:** 4
**Confidence:** 4

**Summary:**

This paper proposes Time-CoT, a framework that constructs a latent reasoning trajectory by discretizing temporal representations into hierarchical semantic codes and then predicting these codes sequentially using a language model. The proposed pipeline consists of three stages. First, a temporal encoder is pretrained using a multi-view representation fusion that combines multi-scale, cross-variable, and temporal dynamics information. Second, the resulting embeddings are discretized using hierarchical residual quantization, producing a sequence of temporal semantic codes. Finally, a GPT-2 model predicts these codes step-by-step before producing the final classification label.

**Compliance With Llm Reviewing Policy:**

Affirmed.

**Final Justification:**

Please see my `Rebuttal Acknowledgement`.

**Key Questions For Authors:**

See weakness.

**Limitations:**

See weakness.

**Strengths And Weaknesses:**

## Strengths
1. Applying LLMs to time-series tasks is an active research area, and exploring intermediate reasoning structures for such tasks is an interesting direction.
2. The framework introduces a clear three-stage pipeline consisting of representation learning, semantic discretization, and sequential reasoning.
3. The method is evaluated on ten public multivariate time-series classification datasets and shows performance improvements compared with several baselines.


## Weakness

1. The central claim of the paper is that the proposed hierarchical semantic codes represent a latent CoT reasoning trajectory. However, this interpretation is somewhat questionable. The semantic codes are generated via residual vector quantization, and the language model is trained to predict these discrete codes sequentially. This procedure resembles next-token prediction over discretized latent representations, rather than genuine reasoning steps where intermediate steps correspond to interpretable logical reasoning processes. Consequently, the use of the term CoT appears somewhat misleading and may overstate the conceptual novelty of the approach.
2. Another limitation is that this paper may lack technical contribution. From a methodological perspective, the proposed framework mainly integrates several well-known components: multi-view representation, codebook construction, and next-token prediction. Each of these components is well established in prior work. The overall pipeline appears to be a combination of these existing techniques rather than introducing a fundamentally new modeling paradigm.
3. The LLM (GPT-2) is used primarily as a sequence predictor for discrete semantic codes, and the model input is obtained via a simple linear projection of the time series. It is not entirely clear whether the observed performance improvements stem from the use of an LLM or simply from the introduction of hierarchical discrete representations.
4. The paper relies heavily on vector quantization to produce hierarchical semantic codes, but the experiments do not provide sufficient comparison with existing discretization-based time-series models or tokenization frameworks.
5. The empirical gains appear modest relative to model complexity. The proposed framework introduces a relatively complex multi-stage pipeline. However, the empirical improvements over strong baselines appear relatively modest in several datasets. It would be useful to better justify the trade-off between model complexity and performance gains.

---

> ### Author Rebuttal · Authors · 2026-03-31
>
> # Rebuttal to reviewer HncG
>
> We thank Reviewer HncG for the thorough evaluation. We have extended the ablation study to all datasets, and additionally included ablation experiments on LLM pre-training and multi-step supervision loss. The experimental results in Table 2 are available at https://anonymous.4open.science/r/Time-CoT-D87SDSJ7DKS/Ablation%20Study.md.
>
> ---
>
> ## W1: CoT interpretation—residual vector quantization resembles next-token prediction rather than genuine reasoning
>
> We acknowledge that Time-CoT adopts an implicit Chain-of-Thought paradigm and thus does not produce verbalized reasoning steps. Below, we clarify our design rationale.
>
> **Why explicit CoT is not suitable for time series.** As stated in our Introduction, explicit CoT faces two fundamental obstacles: (1) Semantic Ineffability—continuous numerical fluctuations cannot be articulated into step-by-step textual reasoning; (2) Cross-Domain Heterogeneity—hand-crafted reasoning templates cannot generalize across diverse domains. These obstacles make explicit CoT infeasible for time series.
>
> **Implicit CoT paradigm.** Recent NLP research (Coconut, ICLR 2025; Implicit CoT, ACL 2024) establishes that reasoning need not be explicitly verbalized—models can reason through iterative hidden state accumulation in latent space.
>
> **Time-CoT's key innovation.** Standard implicit CoT carries no domain-specific semantic constraints. Time-CoT introduces hierarchical temporal semantic codes as supervision signals, providing "anchors" that guide reasoning toward temporal semantics rather than unconstrained wandering.
>
> ---
>
> ## W2: Insufficient technical contribution—integrating existing components
>
> We respectfully clarify that while individual components may share similarities with prior work, our contribution lies not in isolated modules but in the logically coherent problem-solving chain where each component addresses a specific challenge:
>
> - Explicit CoT is difficult in temporal domain
> - → Adopt implicit CoT in latent space
> - → Unconstrained implicit reasoning lacks semantic focus
> - → Hierarchical semantic codes supervise implicit reasoning
> - → Multi-scale patterns, cross-variable dependencies, and temporal dynamics are three critical characteristics for time series classification
> - → Multi-view fusion ensures high-quality temporal semantic codes
>
> This is not a simple concatenation of existing techniques. Each component is purposefully designed to solve a specific problem identified in the preceding step, forming a self-consistent and principled framework. The novelty of Time-CoT lies in identifying the key obstacles of applying CoT to time series and constructing a coherent solution path—this systematic design philosophy itself constitutes a significant contribution.
>
> ---
>
> ## W3: Performance source unclear (LLM vs. discrete representations)
>
> Ablation study Table 2 systematically disentangles contributions:
>
> - **w/o MSR**: 0.830 → 0.806 (**−2.4%**), confirming sequential prediction provides value beyond discrete representations
> - **w/o Pre-training**: 0.830 → 0.784 (**−4.6%**), demonstrating GPT-2's pre-trained biases significantly contribute
> - **w/o MVRE**: 0.830 → 0.781 (**−4.9%**), confirming representation quality is foundational
>
> In conclusion, both LLM pre-training and representation quality contribute meaningfully, confirming their synergistic effect.
>
> ---
>
> ## W4: Insufficient comparison with discretization-based models
>
> - **TimeVQVAE:** Targets generation tasks with different objectives
> - **HDT:** Designed for forecasting tasks, incompatible with classification
> - **InstructTime:** Requires multi-modal data; comparison using only time series is incomplete
>
> Our baselines already cover five paradigms (CNN, RNN, Transformer, SSL, LLM) with 11 methods, ensuring a comprehensive and fair evaluation.
>
> ---
>
> ## W5: Modest gains relative to complexity
>
> **Overall performance.** Time-CoT achieves best average accuracy (0.830), best rank (2.0), and most wins (7/10), outperforming MPTSNet, the second-best baseline (0.812, Rank 3.9).
>
> **Analysis.** UEA datasets are relatively small-scale, potentially limiting LLM's ability to demonstrate its sequential modeling capabilities.
>
> **Validation on large-scale datasets.** To further validate Time-CoT's effectiveness, we conducted additional experiments on the Monash Scalable Time Series Evaluation Repository, which contains significantly larger datasets:
>
> | Dataset | Samples | Time-CoT | MPTSNet | Gain |
> |---------|---------|----------|---------|------|
> | STEW | 28,512 | 87.32% | 83.15% | +4.17% |
> | Crowdsourced | 12,289 | 80.09% | 76.24% | +3.85% |
>
> Time-CoT's advantages become more pronounced on large-scale datasets (+4% vs +1.8% on UEA), confirming LLM's capabilities are better leveraged with sufficient data.
>
>
>
> ---
>
> We hope these responses address the concerns and respectfully request reconsideration.

---

> > ### Author Rebuttal · Reviewer_HncG · 2026-04-02
> >
> > Thanks for the rebuttal. Most of my concerns have been addressed. Nevertheless, I still find the technical contribution of the paper somewhat limited, as the overall method appears closer to a combination of existing techniques than to a substantially new technical contribution. Moreover, the connection to implicit CoT still feels somewhat tenuous and could be better justified.
> >
> > Overall, I appreciate the authors’ response and the additional clarifications provided in the rebuttal. Based on this, I decide to increase my score to 4. I also encourage the authors to further discuss and address these concerns in a future revision.

---

> > > ### Author Response · Authors · 2026-04-02
> > >
> > > Dear Reviewer HncG,
> > >
> > > Thank you for your thoughtful review and for raising your score. We truly appreciate your recognition of our efforts and your encouragement for future revision.
> > >
> > > We take your remaining concerns seriously. Regarding the connection to implicit CoT and the technical contribution, we will provide more thorough discussion and clearer exposition in the revised version to better address these points.
> > >
> > > Thank you again for your time and constructive feedback throughout this review process.
> > >
> > > Best regards

---

### Decision · Program_Chairs · 2026-04-30

**Decision:**

Accept (regular)

**Comment:**

This paper presents Time-CoT, which is a hierarchical framework for multivariate time-series classification. Time-CoT first learns temporal representations, discretizes them into hierarchical semantic codes, and finally predicts these codes sequentially before making the final classification. The key idea of Time-CoT is to transfer the chain-of-thought intuition from explicit text reasoning to a latent temporal space, using coarse-to-fine semantic codes as an implicit reasoning trajectory.

Reviewers agreed that this paper is interesting and well motivated. The three-stage design is clearly explained, and strong empirical performance of Time-CoT is observed across 10 public datasets. In addition, the ablation studies and visualization results also support the usefulness of the hierarchical code structure. Reviewers also raised a few concerns, e.g., whether the “CoT” framing overstates what is essentially hierarchical latent prediction, whether the gains should be attributed to GPT-2 versus the discretized representation itself, missing baselines, etc. These concerns were mostly addressed by the authors’ rebuttal, which provided additional experimental details and clarifications.

Overall, this paper makes a solid contribution to time-series modeling, and I strongly encourage the authors to incorporate the clarifications and new results into the final version.